# SciDA: Scientific Dynamic Assessor of LLMs

## Abstract

Advancement in Large Language Models (LLMs) reasoning capabilities enables them to solve scientific problems with enhanced efficacy. Thereby, a high-quality benchmark for comprehensive and appropriate assessment holds significance, while existing ones either confront the risk of data contamination or lack involved disciplines. To be specific, due to the data source overlap of LLMs training and static benchmark, the keys or number pattern of answers inadvertently memorized (i.e. data contamination), leading to systematic overestimation of their reasoning capabilities, especially numerical reasoning.

We propose **SciDA**, a multidisciplinary benchmark that consists exclusively of over 1k Olympic-level numerical computation problems, allowing randomized numerical initializations for each inference round to avoid reliance on fixed numerical patterns. We conduct a series of experiments with both closed-source and open-source top-performing LLMs, and it is observed that the performance of LLMs drop significantly under random numerical initialization. Thus, we provide a methodology for more truthful and unbiased assessments of LLMs. The evaluation framework has been anonymized and is publicly available at https://anonymous.4open.science/r/SciDA-0184.

## 1 Introduction

Recent advancements in large language models (LLMs) have demonstrated remarkable capabilities in complex reasoning tasks. However, evaluating their true reasoning capabilities remains challenging, particularly in scientific domains, which require multi-step calculation and symbolic manipulation.

To assess the problem-solving capabilities of LLMs quantitatively, a series of benchmarks (GSM8k, MATH, MMLU, etc.) have been created and widely applied (Cobbe et al., 2021; Hendrycks et al., 2021b;a), with data primarily sourced from academic competition questions and textbooks. These early-stage benchmarks have become relatively easy for frontier-level LLMs. Further works (GPQA, MMLU-Pro, Agieval, Scibench, Scieval, etc.) enable more comprehensive and rigorous assessment by incorporating more knowledge domains and diversifying data sources (Rein et al., 2024; Wang et al., 2024; Zhong et al., 2023; Wang et al., 2023; Sun et al., 2024). However, there has been an obscured essential contradiction: open-access textbooks, examination questions, academic literature, and online datasets, which are the primary data sources of benchmarks, also serve as the data sources of LLMs' pretraining and fine-tuning. As a result, data leakage and contamination are highly probable, and certain combinations of numbers can be memorized, thereby hindering their ability to generalize or leading to a systematic overestimation of their cognitive reasoning capabilities (Golchin & Surdeanu, 2023; Deng et al., 2023; Dong et al., 2024). This is particularly concerning in domains requiring numerical reasoning, where reliance on memorized answers or combinations of numbers could have a more pronounced influence.

Generative benchmarks like KORgym(Shi et al., 2025) have been released recently. However, those works focus on game-based interactions, linguistic adaptability, or toy problems, lacking comprehensiveness and scientific rigor. In spite of mathematics, existing benchmarks lack assessment across various branches of natural science (physics, chemistry, biology, etc.), while such capabilities are crucial for real-world applications in scientific research. This gap underscores the need for a comprehensive multi-disciplines dynamically generated benchmark to truthfully reflect the complexity and unpredictability of scientific problem-solving.

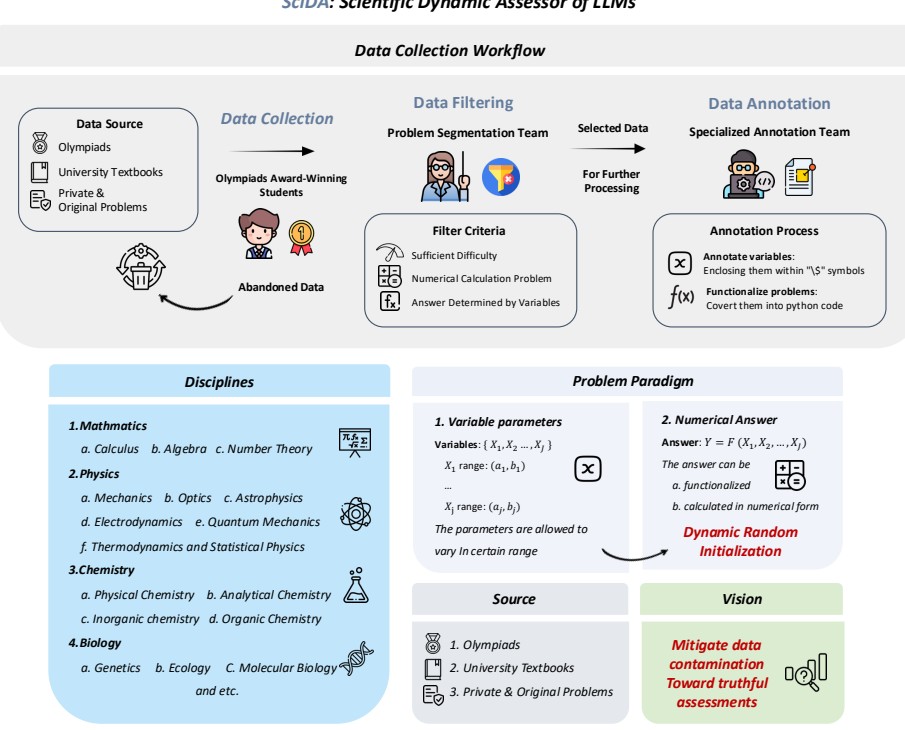

Figure 1: The data construction pipeline of the SciDA. ***Data Collection Workflow*** illustrates the process of collecting and filtering scientific problems from Olympiad competitions and university textbooks, followed by variable annotation and numerical functionalization. ***Discipline*** shows that our benchmark covers various subjects include mathematics, physics, chemistry, and biology. ***Problem Paradigm*** shows that the benchmark supports dynamic random initialization and aims to provide a robust and contamination-free evaluation for scientific reasoning models.

To address these limitations, we introduce **SciDA**, a dynamic scientific benchmark built on 1,000 expert-curated problems from Olympiad-level competitions spanning mathematics, physics, chemistry, and biology. Each problem undergoes structured variable extraction, where all modifiable parameters are programmatically identified and replaced with $ tokens (e.g., $m$, $k$, $v_0$). During each evaluation iteration, these tokens are dynamically initialized with randomized values sampled from predefined scientifically valid ranges. Our data collection pipeline prioritizes quality, diversity, and complexity through domain-expert annotation, symbolic consistency verification, range validation for randomized variables, and solvability checks across value permutations.

To our best knowledge, SciDA is the first dynamic, contamination-proof benchmark for rigorous scientific reasoning evaluation. Our work yields three pivotal insights:

## 2 RELATED WORK

### 2.1 SCIENTIFIC BENCHMARKS

To comprehensively evaluate the performance of current LLMs, series of benchmarks (GSM8k, MATH, MMLU, etc.) have been created (Cobbe et al., 2021; Hendrycks et al., 2021b;a), with data primarily sourced from academic competition questions and textbooks. Further works (GPQA, SuperGPQA, MMLU-Pro, Agieval, Scibench, Scieval, etc.) enable more comprehensive and rigorous assessment by incorporating more disciplines and diversifying data sources (Rein et al., 2024; Team et al., 2025; Wang et al., 2024; Zhong et al., 2023; Wang et al., 2023; Sun et al., 2024). However, owing to the advancement in LLMs capabilities, existing benchmarks have become relatively easy for

advanced LLMs and existing benchmarks. Therefore, the need for scientific benchmarks to assess the limits of advanced LLMs naturally emerged.

Driven by such need, a major focus is to collect problems with higher complexity, primarily Olympics problems, i.e. "the pearl of human wisdom". Undergraduate-level (Liu et al., 2024; Tang et al., 2024) and Olympic-level (He et al., 2024; Huang et al., 2025b; Sun et al., 2025) benchmarks are created and applied. For instance, OlymMATH(Sun et al., 2025) is a benchmark Olympics-level mathematical problems spanning multiple mathematical domains, including algebra and geometry. Omni-MATH(Gao et al., 2024) is also a mathematical benchmark at the Olympic level integrated with a data leakage detection mechanism, on which the most advanced LLMs (such as OpenAI's o1) achieve accuracy rates of only 52.55%.

Another focus is to expand the scope of disciplines and go beyond mathematics. Both multidisciplinary (Wang et al., 2024; 2023; Sun et al., 2024; Huang et al., 2025b) and discipline specific benchmarks(Jain et al., 2025; Qiu et al., 2025) emerge. For instance, SciEvalSun et al., 2024 includes disciplines of physics, chemistry and biology and OlympicArena (Huang et al., 2025b) spans 7 core disciplines: mathematics, physics, chemistry, biology, geography, astronomy, and computer science. Meanwhile, some benchmarks feature domain specific and focus on relatively naive disciplines, such as PHYbench (Qiu et al., 2025) that consists of 500 original physics problems, which fills the blank of high-quality physics benchmarks.

## 2.2 DYNAMIC BENCHMARK

To mitigate data contamination, some researchers update parameters manually and periodically. That is, to create dynamic benchmarks, such as VarBench (Qian et al., 2024) and LiveCodeBench (Jain et al., 2025)), of which the parameters are variable rather than constant. However, such dynamism is artificially maintained pseudo-dynamism. Live updates come with the burden of sustained collection and processing of high-quality data. Thus, the solution is expedient, evading the essential issue.

Further works like KORgym(Shi et al., 2025) do realize dynamic initialization, but they primarily focus on game-based interactions, linguistic adaptability, or toy problems, lacking rigor and not being able to accurately assess the capabilities of scientific problem-solving. Meanwhile, benchmarks like Math-perturb (Huang et al., 2025a), which concerns purely mathematics, lack comprehensiveness and are limited to few discplines. Comprehensive assessment across various branches of natural science (physics, chemistry, biology, etc.) is vital, since such capabilities are crucial for real-world applications in scientific research.

Despite the effectiveness in mitigating data contamination, existing dynamics benchmarks are inconsistent and unsatisfactory in form and quality, while remaining limited to few disciplines, primarily mathematics.

## 3 APPROACH

### 3.1 PROBLEM FORMALIZATION

Let $q \sim \mathcal{Q}$ denote a problem sampled from the problem distribution $\mathcal{Q}$.

Suppose $Q$ contains $J$ random variables, indexed by $i = 1, 2, \ldots, J$. Denote these random variables by

$$\{ \mathcal{X}_1, \mathcal{X}_2, \ldots, \mathcal{X}_J \}.$$

Each $\mathcal{X}_i$ is drawn from a uniform distribution over the interval $[\, a, b \,]$:

$$\mathcal{X}_i \sim \mathcal{U}(a, b),$$

where $a$ is the minimum possible value and $b$ is the maximum possible value, determined by the actual meaning of each variable.

**Initialization.** When initializing the problem, each random variable $\mathcal{X}_i$ is independently sampled to obtain a realization $x_i$:

$$x_i \sim \mathcal{U}(a, b), \quad \forall i \in \{1, 2, \ldots, J\}.$$

Therefore, all variables in problem $q$ are randomly initialized before reasoning.

**Answer generation.** After the initialization of $\{x_1, x_2, \ldots, x_J\}$, the correct answer $\mathbf{y}$ to problem $Q$ is designed to be a finite sequence of real numbers. Formally, there exists a known, labeled function

$$F : \mathbb{R}^J \longrightarrow \mathbb{R}^K,$$

such that

$$\mathbf{y} = F(x_1, x_2, \ldots, x_J)$$

is the groundtruth answer vector of length $K$. Here,

$$\mathbf{y} = (y_1, y_2, \ldots, y_K) \in \mathbb{R}^K$$

depends deterministically on the initialized values $(x_1, x_2, \ldots, x_J)$.

**Model Prediction and Correctness Criterion.** Let $\widehat{\mathbf{y}} = (\hat{y}_1, \hat{y}_2, \ldots, \hat{y}_K)$ denote the sequence of numbers predicted by the model (or solver) in response to $Q$. We say that the models answer is *correct* if its deviation from the true answer $\mathbf{y}$ is within a prescribed tolerance.

### 3.2 DATA COLLECTION

Our data collection process involved three main steps to create a high-quality dataset of variable-based computational problems.

First, a team of students with competition backgrounds meticulously collected problems from regional and international Olympiad competition problems, Olympiad workbook and guides, related online platforms and university textbooks. This broad collection ensured a diverse initial pool of potential problems.

Next, a problem-segmentation team filtered this initial pool based on three strict criteria: 1) sufficient difficulty, 2) being a computational problem containing variables, and 3) the answer being determined by variables and presented in a numerical format. Problems satisfying these criteria were extracted for the next stage.

**Example of SciDA Questions**

**Instruction**
Find the area enclosed by the astroid defined by the parametric equations $x = a\cos^3 t$ and $y = b\sin^3 t$, where $t$ ranges from 0 to $2\pi$. Express the result in terms of $a=\$a\$$, $b=\$b\$$, and fundamental constants.

**A**

**B**

**Arguments:**
name: a
    description: semi-axis length along the x-axis;
    type: float; range: (0,inf);\n
name: b;
    description: semi-axis length along the y-axis;
    type: float; range: (0,inf);

**Python function**
```python
import math
from scipy.integrate
import quad
def calculate_astroid_area(a, b):
    def integrand(t):
        term1 = (math.cos(t)**4) * (math.sin(t)**2)
        term2 = (math.cos(t)**2) * (math.sin(t)**4)
        return term1 + term2

    integral, _ = quad(integrand, 0, 2 * math.pi)
    area = (3 * a * b / 2) * integral
    return round(area, 5)
```
**C**

Figure 2: Example of a **SciDA** problem. (A) shows the problem instruction with variables annotated using "$" symbols. (B) shows how the arguments are labeled. (C) shows the Python code to generate the answer.

Finally, a specialized annotation team took the extracted problems and performed two key tasks: annotating the variables by enclosing them within "$" symbols (as exemplified in Figure 2, and writing Python code to solve each problem.

After completing the annotation, we subjected the variables in our problems to five rounds of initialization. This involved assigning different values to the variables and verifying that the corresponding Python code executed correctly and produced valid results. Following this rigorous cleaning process, we obtained a dataset of 1000 problems that met all our specified criteria. The disciplinary and difficulty distributions of this dataset are presented in Figure 6a. Statistics on the length of the questions are shown in Figure 6b.

## 4 EXPERIMENTS

We have selected and assessed mainstream large language models (LLMs) and on the proposed **SciDA** benchmark with both initial parameters and 5 sets of randomized parameters 5 times (the reason why we chose this hyperparameter is described in B). and selected 14 mainstream models to conduct the experiments. The performance of each model is shown in Table 2.

### 4.1 OVERALL COMPARISON OF MODELS

Generally, the accuracy of various models on our benchmark with initial parameters ranges from 20% to 50%, demonstrating that our benchmark is sufficiently challenging and the problems selected are of satisfactory quality. Under random initialization, the accuracy drop of those models ranges from 10% to 20%, which indicates a significant decrease of 20% to 60% relatively.

Among the assessed models, the performance exhibits significant variation while the top three performers are *Gemini-2.5-pro*, *OpenAI-o3*, and *Doubao1.5-pro-thinking*. They have achieve the highest average accuracy, reaching nearly 50% under the fixed-parameter setting and remaining above 35% under randomization.

Table 1: Model Performance on SciDA

| Model Name | Avg | Initial | | | | Random | | | |
|---|---|---|---|---|---|---|---|---|---|
| | | All | Easy | Medium | Hard | All | Easy | Medium | Hard |
| Gemini-2.5-pro.preview.0506.google.ci | **40.19** | 49.84 | 55.68 | 48.68 | **42.54** | **38.26** | 42.05 | **38.04** | **32.63** |
| OpenAI-o3-high.code | 39.45 | **52.22** | 62.60 | **51.06** | 37.72 | 36.90 | 45.10 | 33.23 | 30.00 |
| Doubao1.5-pro-thinking.0415 | 37.64 | 51.50 | **64.27** | 47.88 | 37.28 | 34.87 | **45.43** | 30.32 | 25.70 |
| DeepSeek-reasoner-R1.volc | 35.37 | 47.98 | 59.28 | 44.18 | 36.40 | 32.84 | 41.44 | 28.78 | 25.96 |
| OpenAI-o4-mini.high.0416.code | 33.18 | 44.47 | 55.12 | 40.74 | 33.77 | 30.92 | 39.89 | 25.93 | 25.00 |
| Qwen3-235b-instruct-2507 | 32.38 | 44.46 | 58.45 | 41.53 | 27.19 | 29.96 | 39.72 | 25.71 | 21.57 |
| Claude-gcp.37.thinking | 32.09 | 48.29 | 63.43 | 44.18 | 31.14 | 28.85 | 39.45 | 24.13 | 19.91 |
| Gemini-2.5-pro.preview.0506 | 30.20 | 46.12 | 53.19 | 46.30 | 34.65 | 27.01 | 32.08 | 24.66 | 22.89 |
| OpenAI-o1-1217.high.code | 29.21 | 41.78 | 51.25 | 39.95 | 29.82 | 26.70 | 34.46 | 22.49 | 21.40 |
| Qwen3-80b-thinking | 27.73 | 40.33 | 48.19 | 38.35 | 31.14 | 25.21 | 30.80 | 23.33 | 19.47 |
| DeepSeek-V3-0324.volc.forCompetitor | 26.65 | 41.99 | 54.85 | 38.10 | 28.07 | 23.58 | 33.46 | 18.73 | 15.96 |
| Kimi-K2 | 25.87 | 41.46 | 56.78 | 34.39 | 28.97 | 22.75 | 32.13 | 18.20 | 15.43 |
| OpenAI-o3-mini.high.code | 25.58 | 42.30 | 52.91 | 37.83 | 32.89 | 22.23 | 27.98 | 18.20 | 19.82 |
| Gemini-2.5-flash.preview.0520 | 24.20 | 41.26 | 51.25 | 38.62 | 29.82 | 20.79 | 27.31 | 17.41 | 16.05 |
| GPT4o-1120 | 14.82 | 28.54 | 40.99 | 25.40 | 14.04 | 12.08 | 18.39 | 9.74 | 5.96 |
| GPT4o-0513 | 14.68 | 27.51 | 40.72 | 23.28 | 13.60 | 12.12 | 19.11 | 10.00 | 4.56 |
| GPT4o-0806 | 12.96 | 25.75 | 40.17 | 20.11 | 12.28 | 10.40 | 16.84 | 7.88 | 4.39 |

Note: Scores represent performance percentages on the **SciDA** benchmark. Initial denotes fixed-parameter problems, while Random denotes dynamically parameterized problems.

#### 4.1.1 GAP BETWEEN REASONING MODELS AND INSTRUCT MODELS

Reasoning-oriented models outperform instruct models siganificantly when dealing with our selected long-chain reasoning numerical problems. More specifically, Gemini-2.5-pro, OpenAI-o3 and Doubao1.5-pro-thinking have superior performance. We attribute this to their enhanced capacity for slow thinking, exemplified by the application of chain-of-thought reasoning, multi-step inference, and emphasis on logical coherence. This observation underscores the necessity of slow thinking and, by extension, highlights the importance of utilizing high-quality, curated, and logically robust training data during the training process, which facilitates models' acquisition of "slow thinking" strategies.

#### 4.1.2 CLOSED-SOURCE MODELS VS. OPEN-SOURCE MODELS

Certain performance gap emerges between closed-source and open-source families. While leading closed-source models surpass 40% accuracy, open-source competitors (*DeepSeek-R1* and *DeepSeek-V3*) varies between 25% to 35%. This discrepancy highlights the non-trivial advantage of exclusive training pipelines.

### 4.1.3 SCALING WITH MODEL SIZE

The results also reveal a positive correlation between model size or generation and their performance. Within each family, newer and larger variants (e.g. OpenAI-o3 vs. OpenAI-o1) exhibit clear gains, particularly on medium and hard problems. However, diminishing returns are observable: performance improvements shrink as models approach the 40% to 50% range, suggesting that further scaling may not be sufficient without dedicated reasoning optimization.

## 4.2 BENCHMARK DISCRIMINABILITY

Our proposed **SciDA** benchmark demonstrates impressive discriminability. On one hand, the gap between the best and weakest models exceeds 30%, indicating that SciDA has proper complexity to discriminate model capabilities. On the other hand, the Easy/Medium/Hard levels are well separated while the accuracy decreases by approximately 8% per level form Eazy to Hard.

Table 2: Model Performance Comperision between SciDA, HLE, and SuperGPQA

| Model Name | SciDA | HLE (Text Only) | SuperGPQA |
|---|---|---|---|
| Gemini-2.5-pro.preview.0506.google.ci | 40.19 | 18.38 | – |
| OpenAI-o3-high.code | 39.45 | 20.57 | – |
| Doubao1.5-pro-thinking.0415 | 37.64 | – | 55.09 |
| DeepSeek-reasoner-R1.volc | 35.37 | 8.5 | 61.82 |
| OpenAI-o4-mini.high.0416.code | 33.18 | 18.08 | – |
| OpenAI-o1 | 29.21 | 8.0 | 60.24 |
| DeepSeek-V3-0324.volc.forCompetitor | 26.65 | 4.55 | 47.40 |
| OpenAI-o3-mini (High) | 25.58 | 13.4 | 55.52 |
| Gemini-2.5-flash.preview.0520 | 24.20 | 12.58 | – |
| GPT4o-1120 | 14.82 | 2.7 | 44.40 |
| Kendalls $\tau$ with SciDA | – | 0.56 | 0.47 |

To further validate the reliability of **SciDA**, we investigate its consistency with two well-recognized benchmarks, **HLE** and **SuperGPQA**, the former featuring the depth of knowledge and the latter the diversity of domains. Calculation details can be found at D

Based on the ranking of the performance of overlapping models, we calculate that correlation between SciDA and SuperGPQA is $\tau = 0.47$, while that between SciDA and HLE is $\tau = 0.56$. Both values indicate a positive correlation, with the stronger alignment observed between SciDA and HLE. We believe such observation can be attributed to the shared characteristics of these two benchmarks, which is their broad domain coverage and emphasis on the difficulty of individual problems.

The overall consistency with established benchmarks n terms of effectiveness suggests the SciDA is fully capable as a robust assessor of LLM's reasoning capabilities.

## 4.3 SUBJECT-SPECIFIC OBSERVATIONS

We further observe that LLMs have a biased performance across different subjects. Performance degradation under randomization is most pronounced in mathematics and physics, with accuracy drops ranging from 30% to 70%. Chemistry and biology, in contrast, exhibit smaller declines (below 50%).

This suggests that subjects requiring longer chains of thought (CoT) and involve more variables (mathematics, physics) are particularly sensitive to parameter perturbations, while fact-based subjects shows more stability. As a result, more obvious deviations in accuracy are observed in mathematics and physics. This aligns with the design goal of **SciDA**, which is to emphasize reasoning over memorization, thereby revealing true differences in model capabilities.

Table 3: Different Model Performance Data Summary by Model, Subject, Type, and Difficulty

| Model Name | Subject | Avg | Initial | | | | Random | | | |
|---|---|---|---|---|---|---|---|---|---|---|
| | | | All | Easy | Med | Hard | All | Easy | Med | Hard |
| OpenAI-o3-high.code | Biology | 55.10 | 82.31 | 82.47 | 85.00 | 70.00 | 49.66 | 53.20 | 42.00 | 46.00 |
| | Chemistry | 35.37 | 48.98 | 61.22 | 46.94 | 38.78 | 32.65 | 35.51 | 36.33 | 26.12 |
| | Physics | 29.84 | 37.43 | 48.74 | 34.97 | 23.61 | 28.32 | 37.14 | 25.17 | 20.00 |
| | Math | 43.90 | 55.16 | 60.42 | 58.90 | 44.33 | 41.65 | 51.67 | 37.67 | 37.73 |
| Doubao1.5-pro-thinking.0415 | Biology | 56.35 | 82.31 | 83.51 | 87.50 | 50.00 | 51.16 | 56.29 | 41.00 | 42.00 |
| | Chemistry | 35.26 | 48.98 | 55.10 | 48.98 | 42.86 | 32.52 | 38.37 | 33.47 | 25.71 |
| | Physics | 29.44 | 38.92 | 54.62 | 33.57 | 23.61 | 27.54 | 39.33 | 22.24 | 18.61 |
| | Math | 38.64 | 51.62 | 61.46 | 50.68 | 43.30 | 36.05 | 45.63 | 34.25 | 29.28 |
| DeepSeek-reasoner-R1.volc | Biology | 53.17 | 74.83 | 72.16 | 85.00 | 60.00 | 48.84 | 51.75 | 41.50 | 50.00 |
| | Chemistry | 34.58 | 41.50 | 53.06 | 34.69 | 36.73 | 33.20 | 35.10 | 34.69 | 29.80 |
| | Physics | 28.24 | 37.13 | 50.42 | 34.27 | 20.83 | 26.47 | 38.15 | 21.40 | 17.22 |
| | Math | 35.00 | 49.85 | 60.42 | 45.89 | 45.36 | 32.04 | 38.33 | 30.55 | 28.04 |
| OpenAI-o4-mini.high.0416.code | Biology | 49.55 | 74.15 | 75.26 | 77.50 | 50.00 | 44.63 | 48.45 | 39.00 | 30.00 |
| | Chemistry | 24.60 | 31.97 | 34.69 | 28.57 | 32.65 | 23.13 | 27.76 | 21.63 | 20.00 |
| | Physics | 23.60 | 32.04 | 43.70 | 27.97 | 20.83 | 21.92 | 31.09 | 17.06 | 16.39 |
| | Math | 39.23 | 49.26 | 59.38 | 47.26 | 42.27 | 37.23 | 48.33 | 32.47 | 33.40 |
| Claude-gcp.37.thinking | Biology | 57.14 | 78.23 | 81.44 | 80.00 | 40.00 | 52.93 | 56.91 | 47.50 | 36.00 |
| | Chemistry | 32.88 | 47.62 | 63.27 | 48.98 | 30.61 | 29.93 | 36.33 | 29.39 | 24.08 |
| | Physics | 27.15 | 39.82 | 50.42 | 35.66 | 30.56 | 24.61 | 35.46 | 19.30 | 17.22 |
| | Math | 25.76 | 43.95 | 61.46 | 41.10 | 30.93 | 22.12 | 28.33 | 20.68 | 18.14 |
| QwenAPI-3-235b-a22b-instruct-2507 | Math | 33.14 | 45.75 | 59.38 | 43.84 | 34.02 | 20.53 | 39.79 | 28.77 | 24.54 |
| | Physics | 27.79 | 35.83 | 52.94 | 32.17 | 20.83 | 19.74 | 37.14 | 21.68 | 15.83 |
| | Chemistry | 28.91 | 32.99 | 44.90 | 34.69 | 18.37 | 24.82 | 35.92 | 25.31 | 23.27 |
| | Biology | 44.56 | 54.66 | 71.13 | 75.00 | 50.00 | 34.45 | 44.74 | 29.50 | 26.00 |
| QwenAPI-3-next-80b-a3b-thinking | Math | 31.42 | 45.40 | 55.21 | 48.63 | 42.27 | 17.43 | 28.96 | 28.36 | 26.39 |
| | Physics | 22.70 | 31.38 | 48.74 | 24.48 | 20.83 | 14.03 | 29.41 | 16.92 | 14.17 |
| | Chemistry | 18.48 | 24.82 | 24.49 | 24.49 | 24.49 | 12.14 | 16.33 | 21.63 | 13.88 |
| | Biology | 39.91 | 47.91 | 52.58 | 67.50 | 30.00 | 31.92 | 41.65 | 30.00 | 18.00 |
| Kimi-K2 | Math | 25.07 | 46.68 | 60.42 | 40.41 | 40.21 | 16.64 | 27.29 | 19.62 | 16.49 |
| | Physics | 18.46 | 27.08 | 47.06 | 20.28 | 13.89 | 9.58 | 27.23 | 10.39 | 10.83 |
| | Chemistry | 22.56 | 31.96 | 44.90 | 26.53 | 22.45 | 13.17 | 25.31 | 20.08 | 17.21 |
| | Biology | 47.85 | 60.16 | 71.13 | 72.50 | 60.00 | 35.53 | 46.58 | 39.00 | 30.61 |
| Gemini-2.5-pro.preview.0506 | Biology | 46.94 | 70.75 | 68.04 | 82.50 | 50.00 | 42.18 | 45.15 | 36.00 | 38.00 |
| | Chemistry | 24.94 | 38.78 | 34.69 | 42.86 | 38.78 | 22.18 | 25.31 | 22.04 | 19.18 |
| | Physics | 24.30 | 38.02 | 45.38 | 38.46 | 25.00 | 21.56 | 26.55 | 18.04 | 20.28 |
| | Math | 31.02 | 46.61 | 57.29 | 45.21 | 38.14 | 27.91 | 29.17 | 28.90 | 25.15 |
| OpenAI-o1-1217.high.code | Biology | 46.49 | 77.55 | 79.38 | 77.50 | 60.00 | 40.27 | 45.57 | 29.50 | 32.00 |
| | Chemistry | 19.84 | 25.17 | 28.57 | 22.45 | 24.49 | 18.78 | 20.41 | 18.37 | 17.55 |
| | Physics | 19.61 | 27.54 | 35.29 | 25.17 | 19.44 | 18.02 | 24.71 | 15.10 | 12.78 |
| | Math | 35.25 | 47.49 | 54.17 | 50.00 | 37.11 | 32.80 | 42.50 | 29.18 | 28.66 |
| DeepSeek-V3-0324.volc.forCompetitor | Biology | 51.36 | 74.15 | 75.26 | 77.50 | 50.00 | 46.80 | 50.10 | 39.50 | 44.00 |
| | Chemistry | 27.55 | 41.50 | 46.94 | 40.82 | 36.73 | 24.76 | 33.47 | 22.04 | 18.78 |
| | Physics | 19.81 | 32.93 | 45.38 | 30.77 | 16.67 | 17.19 | 26.55 | 13.43 | 9.17 |
| | Math | 22.27 | 37.17 | 50.00 | 33.56 | 29.90 | 19.29 | 25.21 | 17.12 | 16.70 |
| OpenAI-o3-mini.high.code | Biology | 41.16 | 72.11 | 72.16 | 75.00 | 60.00 | 34.97 | 39.59 | 23.50 | 36.00 |
| | Chemistry | 17.46 | 23.13 | 30.61 | 18.37 | 20.41 | 16.33 | 17.96 | 14.29 | 16.73 |
| | Physics | 18.46 | 32.04 | 42.86 | 26.57 | 25.00 | 15.75 | 20.17 | 13.29 | 13.33 |
| | Math | 29.35 | 47.79 | 57.29 | 45.21 | 42.27 | 25.66 | 31.04 | 22.88 | 24.54 |
| Gemini-2.5-flash.preview.0520 | Biology | 47.96 | 74.83 | 75.26 | 77.50 | 50.00 | 42.59 | 46.60 | 34.50 | 36.00 |
| | Chemistry | 19.95 | 31.29 | 34.69 | 28.57 | 30.61 | 17.69 | 19.18 | 17.55 | 16.33 |
| | Physics | 18.41 | 31.74 | 42.02 | 27.97 | 22.22 | 15.75 | 21.18 | 12.17 | 13.89 |
| | Math | 21.44 | 40.41 | 46.88 | 41.78 | 31.96 | 17.64 | 19.58 | 17.81 | 15.46 |
| GPT4o-1120 | Biology | 36.17 | 61.90 | 63.92 | 65.00 | 30.00 | 31.02 | 33.20 | 30.00 | 14.00 |
| | Chemistry | 14.51 | 23.81 | 30.61 | 24.49 | 16.33 | 12.65 | 19.59 | 6.12 | 12.24 |
| | Physics | 10.63 | 19.76 | 31.09 | 16.78 | 6.94 | 8.80 | 12.94 | 7.55 | 4.44 |
| | Math | 9.83 | 24.78 | 35.42 | 23.29 | 16.49 | 6.84 | 9.58 | 7.53 | 3.09 |
| GPT4o-0513 | Biology | 36.05 | 59.86 | 61.86 | 62.50 | 30.00 | 31.29 | 33.40 | 30.00 | 16.00 |
| | Chemistry | 15.53 | 22.45 | 30.61 | 22.45 | 14.29 | 14.15 | 23.27 | 11.02 | 8.16 |
| | Physics | 9.88 | 20.06 | 33.61 | 16.08 | 5.56 | 7.84 | 13.11 | 6.43 | 1.94 |
| | Math | 9.78 | 23.01 | 33.33 | 19.86 | 17.53 | 7.14 | 10.00 | 7.67 | 3.51 |
| GPT4o-0806 | Biology | 34.81 | 57.82 | 63.92 | 55.00 | 10.00 | 30.20 | 34.02 | 25.00 | 14.00 |
| | Chemistry | 12.59 | 19.05 | 32.65 | 8.16 | 16.33 | 11.29 | 19.59 | 7.76 | 6.53 |
| | Physics | 8.23 | 19.76 | 33.61 | 14.69 | 6.94 | 5.93 | 9.08 | 4.76 | 3.06 |
| | Math | 8.31 | 20.65 | 28.13 | 19.86 | 14.43 | 5.84 | 7.71 | 6.30 | 3.30 |

# 5 DISCUSSION

## 5.1 REASONING BEHAVIOR ANLAYSIS

To further explore the models performance under different configurations, we analyzed the reasoning chain behavior distribution of 200 identical questions with parameters set to either the initial or ran-

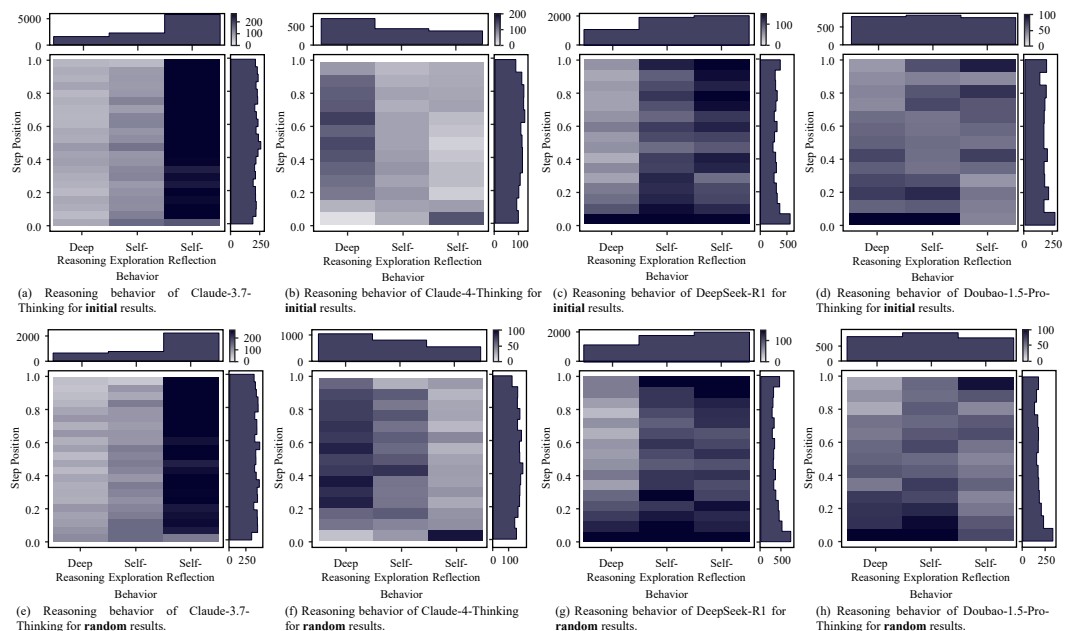

(a) Reasoning behavior of Claude-3.7-Thinking for **initial** results.

(b) Reasoning behavior of Claude-4-Thinking for **initial** results.

(c) Reasoning behavior of DeepSeek-R1 for **initial** results.

(d) Reasoning behavior of Doubao-1.5-Pro-Thinking for **initial** results.

(e) Reasoning behavior of Claude-3.7-Thinking for **random** results.

(f) Reasoning behavior of Claude-4-Thinking for **random** results.

(g) Reasoning behavior of DeepSeek-R1 for **random** results.

(h) Reasoning behavior of Doubao-1.5-Pro-Thinking for **random** results.

Figure 3: The reasoning behavior analysis of initial and random settings.

dom setting using Qwen2.5-32B-Instruct. As shown in Figure 3, the model exhibited a significantly higher frequency of reasoning behaviors in the random setting. This suggests that when presented with more uncertain or complex inputs, the model relies more on reasoning rather than memory, engaging in self-exploration to navigate through multiple potential solutions. The increased reasoning behavior in this setting also includes deeper levels of deep reasoning, where the model performs multi-step logical processing to derive answers. In contrast, the initial setting resulted in fewer reasoning actions, with the model relying more on pre-existing knowledge or memorized patterns, reducing the need for exploration or complex deductions. Therefore, it highlights that the random setting fosters more analytical and proactive problem-solving, requiring the model to engage in more in-depth reasoning, whereas the initial setting leads to quicker, memory-based responses.

## 5.2 TRAINED PROBLEMS CANNOT ROBUSTLY GENERALIZE TO OTHER PROBLEMS WITH THE SAME SOLUTION APPROACH

To further analyze why the model fails to correctly solve problems after initializing the random parameter, we conducted a meticulous manual verification of its incorrect answers. For each subject, we selected one incorrect answer generated by either the thinking model (OpenAI-o3-high.code) or the instruct model (GPT4o-1120) for detailed human scrutiny. Specifically, we randomly sampled 50 problems from each subject (35 in chemistry for GPT4o-1120) and examined all of their corresponding incorrect responses.

We categorized the identified errors based on the following criteria: Logical Errors encompassed issues where the calculation method was incorrect, an erroneous formula was applied, or the reasoning process contained fundamental flaws, indicating the model's failure to grasp the problem or apply appropriate solution strategies. Calculation Errors, on the other hand, included unit confusion, prevalent answer precision problems, incorrect intermediate numerical calculations, or minor computational missteps despite the overall method being correct, suggesting the model struggled with the execution of valid solution steps.

The statistic results are shown in Figure 4. A consistent trend in the distribution of error types was observed across both models. In all disciplines other than Biology, calculation errors were dominant, constituting at least two-thirds of all errors. This indicates that the models were likely trained on larger corpora for these subjects, resulting in superior generalization. Therefore, the main bottleneck appears to be the models' computational capacity. Conversely, for subjects such as biology where

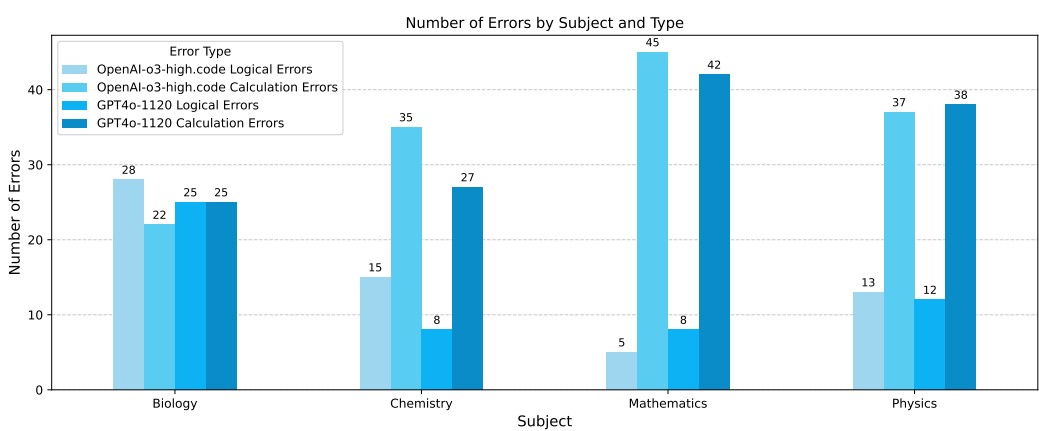

Figure 4: Distribution of error types for different models across various subjects.

corpus is relatively scarce, weaker generalization capabilities likely cause a higher incidence of logical errors, leading them to occur at a frequency nearly equivalent to that of calculation errors.

In summary, we suggest that a model's error types on problems with randomized parameters can, to some extent, reflect its generalization ability in the corresponding discipline. While better generalization leads to fewer logical reasoning errors, challenges with calculation and instruction-following persist. Furthermore, this generalization capability is likely correlated with the richness of the relevant training data.

## 6 CONCLUSION

To conduct comprehensive and truthful assessment of LLMs reasoning capabilities while mitigating data contamination, we proposed **SciDA**, a dynamic multi-disciplinary benchmark. SciDA contains over 1k expert-annotated numerical problems, each of which can be randomized to prevent reliance on memorized numerical patterns. Thus, we ensures that the cognitive reasoning and problem-solving capabilities of LLMs are accurately assessed without bias.

Our experimental results demonstrate that the **even state-of-the-art models reasoning abilities and are far from robust**. While top performers like *Gemini-2.5-pro* and *Doubao1.5-pro-thinking*) reach nearly 50% accuracy under fixed parameters, their performance drops around 15% absolute points under randomization, corresponding to a relative decline of 20% to 60%. This indicates **the vulnerability current LLMs reasoning ability which relies heavily on of reutilization of memorized patterns rather than genuine problem-solving**.

To conclude, our **SciDA exposes the long-exsiting overestimation of LLMs reasoning abilities**, revealing that current models still heavily rely on memorized knowledge and can not truly understanding how to solve complex scientific problems. Our SciDA provides a new paradigm of scientific benchmark allowing dynamical initializations to mitigate data contamination. Moreover, we believe that our work would undoubtedly play a role in narrowing the gap between talented human scientists and LLMs and such advancement towards Artificial General Intelligence could facilitate the possibility of LLMs to advance the frontier of human knowledge.

## 7 FUTURE WORK

We are actively working to expand the scale and disciplinary coverage of SciDA. Our goal is to extend beyond the common STEM subjectssuch as mathematics, physics, chemistry, and biologyto include a wider variety of disciplines. This expansion will enable a more thorough evaluation of LLMs' performance on diverse data, thereby establishing SciDA as a valuable and comprehensive benchmark dataset for the LLM community.

## 8 ETHICS STATEMENT

All datasets used are publicly available and released under appropriate licenses. We carefully considered potential societal impacts of this research. Our models and methods are developed solely for academic research purposes. Potential misuse, such as generating harmful or misleading content, should be mitigated by responsible use and deployment.

## 9 REPRODUCIBILITY STATEMENT

We guarantee reproducibility of our work by fully releasing the datasets, preprocessing scripts, and evaluation framework to the research community. All experimental results can be independently verified using the provided resources. The training and evaluation environment strictly follows the settings of the open-source framework we build upon, ensuring transparency and ease of replication.

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

## A  DATA SOURCE

Our data covers various disciplines and includes both publicly available and privately held or original Olympic-level problems.

We have meticulously sourced problems that meet our requirements from regional and international Olympiad competition problems, Olympiad workbook and guides, and professional college textbooks. The publicly available sources includes:

1. **Mathematics**: International Mathematical Olympiad (IMO), Chinese Mathematical Olympiad (CMO), Problems in Mathematical Analysis by B. P. Demidovich, Euler Math, etc.

2. **Physics**: International Physics Olympiad (IPhO), Chinese Mathematical Olympiad (CPhO), International Physics Olympiad Training and Selection by Yongling Zheng, Collection of Physics Challenges by Yousheng Shu et al., A Grand Dictionary of Plysics Prolens and Solutons by Yongde Zhang et al., New Concept Physics Tutorial by Kaihua Zhao et al., Mechanics by Yousheng Shu et al., etc., Modern Quantum Mechenics by Sakurai Jun, Quantum Mechenics Solution Manual by David J. Griffiths, Electrodynamics Solution Manual by David J. Griffiths, etc.

3. **Chemistry**: International Chemistry Olympiad (IChO), Chinese Chemistry Olympiad (CChO), Physical Chemistry by Peter Atkins, etc.

4. **Biology**: International Biology Olympiad (IBO), Chinese National Biology Olympiad (CNBO), etc.

High-quality privately held or original problems constitute another pillar of our testing benchmark. These problems are contributed by trusted Olympic competition medalists, coaches, and university professors, accounting for over 20% of the total data volume, while the proportion is higher in chemistry and biology.

## B  ELBOW POINT ANALYSIS FOR OPTIMAL RANDOM SAMPLING NUMBER

In our main analysis, we performed 5 random parameter initializations for each problem. Here, we present the elbow point analysis conducted to determine the optimal number of random initializations, denoted as n.

Specifically, we utilized two models: GPT-4o-1120 as the instruction model and OpenAI-o3-mini.high.code as the thinking model. For each model, we conducted 10 independent random parameter initializations and performed inference accordingly. From the 10 resulting sets of outputs for each model, we created subgroups of size n, where n ranged from 2 to 10. For each value of

n, we repeatedly sampled n sets from the 10 sets and calculated the mean of these samples. We then computed the variance of this distribution of means. By plotting the relationship between n and the variance of the means, we identified the "elbow point", which represents the optimal random sampling number.

The results are displayed in Figure 5. As can be seen, n=5 is the elbow point for the inference results of both models. This value represents the most robust and cost-effective choice. Increasing the number of random initializations to 6 or more yields diminishing returns, as the marginal benefit in performance does not justify the additional time and computational costs. Therefore, we conclude that the optimal random sampling number is 5, as it strikes the best balance between resource consumption for inference and the accuracy of the model evaluation.

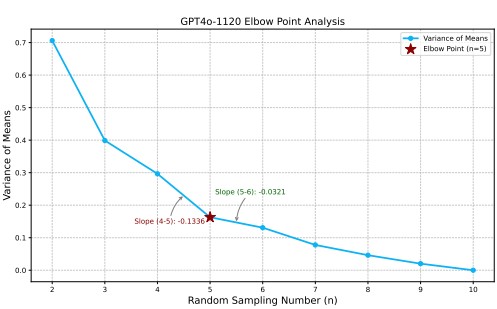

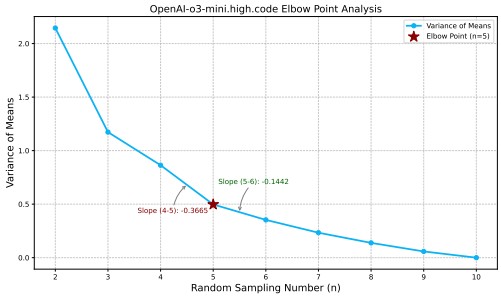

(a) Elbow point analysis for GPT4o-1120 model showing the relationship between the variance of means and the random sampling number (n).

(b) Elbow point analysis for OpenAI-o3-mini.high.code model showing the relationship between the variance of means and the random sampling number (n).

Figure 5: Elbow point analysis for optimal random sampling number.

## C  DIFFICULTY AND LENGTH STATISTICS

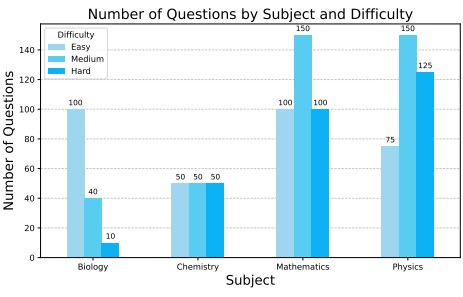

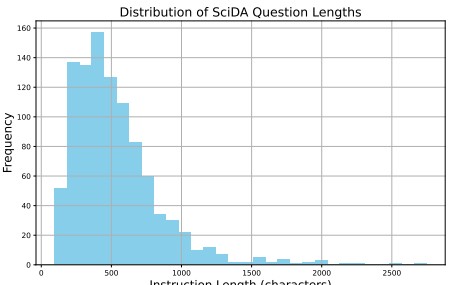

(a) Disciplinary and difficulty distribution of the collected dataset.

(b) Detailed difficulty information for the problems in the dataset.

Figure 6: Distribution and difficulty information of the collected dataset.

## D  KENDALL'S RANK CORRELATION COEFFICIENCY

We adopt Kendalls $\tau$ as the measure of partial order similarity, which evaluates the correlation between two rankings by counting concordant and discordant pairs. Given a set of $n$ models with two rankings, $\tau$ is defined as

$$\tau = \frac{C - D}{\frac{1}{2}n(n - 1)}, \tag{1}$$

where $C$ denotes the number of concordant pairs and $D$ the number of discordant pairs across the two rankings.

## E    USE OF LLMS

Large Language Models (LLMs) were used during the preparation of this paper. Specifically, they were employed for improving the clarity of writing. All experimental design, implementation, and analysis were performed by the authors. No LLM outputs were directly included as research results.

