# OpenReview forum: "SciDA: Scientific Dynamic Assessor of LLMs"
_ICLR.cc/2026/Conference — Submitted to ICLR 2026_

### Official Review · Reviewer_tnzN · 2025-10-24

**Soundness:** 2
**Presentation:** 3
**Contribution:** 2
**Rating:** 4
**Confidence:** 4

**Summary:**

This paper introduces SciDA, a dynamic, multidisciplinary benchmark designed to assess the numerical reasoning abilities of large language models (LLMs) while mitigating data contamination. SciDA consists of over 1,000 expert-curated, Olympic-level numerical computation problems covering mathematics, physics, chemistry, and biology, with dynamic variable initialization to systematically eliminate answer memorization and pattern-based shortcuts. The authors evaluate a wide suite of contemporary LLMs in both fixed and randomized settings and find significant performance drops under randomization, suggesting overestimation of existing models’ reasoning abilities on static benchmarks.

**Strengths:**

1. Dynamic Contamination-Proof Design: SciDA leverages randomized parameter initialization, which directly tackles data contamination and memorization issues commonly plaguing static benchmarks.
2. Multidisciplinary Coverage: By including computation-heavy problems from mathematics, physics, chemistry, and biology, SciDA offers broader evaluative reach than most prior benchmarks limited to single domains.
3. Robust Experimental Evaluation: The authors present comprehensive experiments involving 14 LLMs, reporting per-model results in both fixed and random settings.
4. Insightful Error Analysis: The paper provides a nuanced breakdown of error types and links performance degradation to the lack of true numerical/generalization skills.

**Weaknesses:**

1. Limited Discussion of Theoretical Underpinnings of Randomization Strategy: While Equations (3.1)–(3.5) define the sampling and evaluation framework, the theoretical justification for why uniform randomization over the prescribed intervals is sufficient to prevent contamination (vs. adversarial or structured randomization), or how initialization ranges are determined for each problem, remains underexplored.
2. Potential Annotation and Validation Quality Risks: The annotation/validation process relies on teams of medalists and students, but the degree of independent double-checking, systematic error checking, or inter-annotator agreement is not quantified.
3. Insufficient Mathematical Formalization of Correctness/Tolerances: While the paper specifies that model predictions are deemed correct if "within a prescribed tolerance," the exact metrics are not made explicit, nor are the thresholds justified for each domain/problem type.

**Questions:**

1. Unclear Definition and Justification of Correctness Tolerances Across Domains: How are correctness tolerances defined and justified across the different scientific domains? For example, do relative/absolute error thresholds vary by subject (e.g., chemistry vs. mathematics), and what is the rationale for those choices?
2. Absence of Quantitative Uncertainty and Error Bar Reporting: Are there quantitative uncertainty/error bars (e.g., variance across runs) that could be reported for the benchmark’s headline metrics?

---

> ### Author Response · Authors · 2025-11-17
>
> Distinguished reviewer,
>
> We appreciate your rigorous review and the recognition of our dynamic contamination-proof design, multidisciplinary coverage, and insightful error analysis. We would like to address your concerns below.
>
> ## R1. Randomization Strategy
>
> **Reviewer’s concern**
> The theoretical justification for using uniform randomization over prescribed intervals (vs. adversarial/structured randomization), and how initialization ranges are determined.
>
> **Our response to this concern:**
>
> 1. **Why uniform randomization.**
>    Our goal in this work is robust, contamination-resistant benchmarking, not adversarial robustness. Uniform sampling ensures unbiased coverage of the scientifically valid parameter space without favoring “easy” or “hard” instantiations.
>
> 2. **How ranges are chosen.**
>    1. In Section 3.1–3.2, we formalized each variable \(X_i\) as being drawn from \([a_i, b_i]\), determined by the actual meaning of each variable. The ranges are designed by domain expert annotators and have passed double-check in our data annotation pipeline.
>    2. We conducted validation via multi-round initialization. For each problem, we perform at least five independent random initializations per problem and require that the solver executes successfully and produces valid numerical outputs for all of them, which makes sure the solvability.
>
> ## R2. Annotation and validation quality risks
>
> **Reviewer’s concern**
> The annotation and validation process relies on medalists and students, but the degree of independent double-checking is not quantified.
>
> **Our response to this concern:**
>
> 1. **Multi-stage expert pipeline**
>    Sec. 3.2 describes our pipeline:
>    (i) competition-trained students collect candidate problems;
>    (ii) a problem-segmentation team filters by strict criteria;
>    (iii) an engineer annotation team functionalizes variables and writes Python solvers;
>    (iv) all problems undergo 5 rounds of random initialization with automatic execution checks.
>
> 2. **Independent checking and revision**
>    We admit that we have not clearly specified who is responsible for the independent double-check process. In practice, such work is done by the authors as well as some of the medalists, all of whom have profound knowledge and rigor in their domains. Multiple rounds of checking further enhance reliability.

---

> ### Author Response · Authors · 2025-11-17
>
> ## R3. Mathematical tolerances
>
> **Reviewer’s concern**
> The paper mentions that predictions are considered correct if they are “within a prescribed tolerance,” but the exact metrics and thresholds are not explicitly formalized or justified.
> How are correctness tolerances defined and justified across different scientific domains? Do thresholds vary by subject (e.g., chemistry vs. mathematics), and what is the rationale?
>
> **Our response to this concern:**
>
> Our tolerance is annotated by experts during data processing. Generally, it is **three significant figures or 1%**, which can be slightly different across problems. The tolerance follows two principles:
>
> 1. The tolerance should be tight enough that qualitatively wrong reasoning is penalized.
> 2. Choosing any reasonable solution method or making reasonable approximations in the intermediate steps will not result in penalties.
>
> ## R4. Quantitative uncertainty and error bars
>
> **Reviewer’s concern**
> Are quantitative uncertainty estimates/error bars (e.g., variance across runs) available for the benchmark’s headline metrics?
>
> **Our response to this concern:**
>
> As described in Sec. 4 and App. B, each benchmark score under the “random” setting already aggregates performance over **5 independent random parameter initializations per problem**. App. B presents an elbow-point analysis supporting this choice as a trade-off between stability and cost. We can directly compute the empirical variance (or standard error) of accuracy across these samples for each model, though we did not include these statistics in the current draft due to space limitations. We will add these quantitative uncertainty estimates in the revised version.
>
> Again we thank you for your constructive and detailed feedback. We are glad to offer these clarifications and we hope this response finds you well :)

---

### Official Review · Reviewer_AZaQ · 2025-10-26

**Soundness:** 3
**Presentation:** 3
**Contribution:** 3
**Rating:** 4
**Confidence:** 4

**Summary:**

This paper introduces SciDA, a new dynamic benchmark with over 1,000 scientific problems designed to address performance overestimation in LLMs caused by data contamination in static benchmarks. Its core feature is the random initialization of numerical parameters for each problem, preventing models from relying on memorization. The authors' key finding is that this randomization causes a significant drop in accuracy for all tested LLMs, revealing a vulnerability in their reasoning capabilities.

**Strengths:**

1. The problem this paper addresses is a well-recognized and critical issue in the field of LLM evaluation.
2. The dynamic "functionalization" and random initialization of problems is a sound and necessary methodology for testing true generalization over memorized patterns.
3. The rigorous, expert-led data collection and annotation pipeline ensures a high-quality, difficult set of problems that yield valuable insights into model capabilities.

**Weaknesses:**

1. The dataset's scale (1,000 problems) is limited when distributed across four distinct scientific disciplines and multiple difficulty levels, which may affect the statistical reliability of results in specific sub-domains.

2. The paper's main conclusion is ambiguous. The performance drop is attributed to a failure in "genuine problem-solving" , yet the paper's own error analysis (Figure 4) shows "Calculation Errors" are far more common than "Logical Errors" in most subjects . This suggests models might be generalizing the logic correctly but failing at the computation. If the logic is correct, do the models just need a calculator or code interpreter tool to solve the problems?

3. Key methodological details are missing, specifically how the "scientifically valid ranges" for parameter randomization were defined and validated.

**Questions:**

1.  Could the authors please provide a more detailed statistical breakdown of the 1000 problems in an appendix? Specifically, a table showing the exact distribution across the four disciplines and three difficulty levels (as referenced in Figure 6a 35) would be very helpful.

2.  I suggest the authors refine their claim of being the "first" dynamic benchmark. Please clarify the distinction from KORgym and Math-perturb more explicitly in the Related Work section and adjust the contribution statement to be more precise.

3. Please elaborate on the implications of the findings in Figure 3,4. If models are primarily making "Calculation Errors," does this not imply that their logical generalization is (at least partially) successful, but their internal arithmetic/symbolic manipulation capabilities are brittle? This distinction is crucial for understanding what "reasoning" means for LLMs and where the true bottleneck lies.

---

> ### Author Response · Authors · 2025-11-17
>
> Distinguished reviewer:
>
> We appreciate your rigorous review and the recognition of the significance of this issue, as well as the positive ratings on our methodology and annotation pipeline, and we would like to address the concerns below.
>
> ## R1. Dataset scale and statistical reliability in sub-domains
>
> We agree that, once split across 4 disciplines × 3 difficulty levels, the statistical reliability in corresponding domains can be slightly impacted.
>
> **Our response**
>
> - We value quality and difficulty rather than raw volume. Since the ~1000 problems is expert-annotated, Olympiad-level, We hope that an appropriate number of high-quality questions can clearly demonstrate our insight.
>
> - SciDA yields stable and meaningful trends at different discipline × difficulty levels:
>   1. Clear separation between Easy/Medium/Hard with accuracy drop that makes sense (Table 1 and Table 3).
>   2. Subject-specific patterns (e.g., stronger degradation under randomization in mathematics/physics) that are consistent across several top models (Sec. 4.3).
>
> - **Further Statistical reporting.**
>   We agree that discussion on standard errors / confidence intervals will add credibility, and it will be prioritized in our further revision.
>
> ## R2. Ambiguity of the main conclusion; “genuine problem-solving” vs. calculation errors
>
> **Our response**
>
> 1. **What SciDA actually reveals.**
>    We will refine the wording to avoid suggesting that models fail at all aspects of reasoning. In our view, “genuine scientific problem-solving” requires the joint reliability of:
>    - (a) selecting an appropriate solution strategy and formulas;
>    - (b) correctly manipulating those formulas;
>    - (c) executing multi-step numerical calculations under randomized parameters.
>
>    SciDA reveals that, even when (a) is often correct, (b)–(c) are fragile: small perturbations in parameters substantially increase the chance of numerical mistakes and instruction-following failures.
>
> 2. **Tool Uasge.**
>    We agree that tool support (calculators, code interpreters) could close part of the gap. Our current experiments intentionally evaluate “pure” LLM reasoning without tools. We do believe both the core reasoning capability of pure LLM and tool invocation  has considerable significance.
>
> ## R3. Details for “scientifically valid ranges”
>
> **Reviewer’s concern.**
> The paper does not clearly explain how the “scientifically valid ranges” for parameter randomization are defined and validated.
>
> **Our response to this concern:**
>
> 1. In Section 3.1–3.2, we formalized each variable \(X_i\) as being drawn from \([a_i, b_i]\), determined by the actual meaning of each variable. The ranges are designed by domain expert annotators and have passed double-check in our data annotation pipeline.
>
> 2. We conducted validation via multi-round initialization. For each problem, we perform at least five independent random initializations and require that the solver executes successfully and produces valid numerical outputs for all of them, which makes sure the solvability.

---

> ### Author Response · Authors · 2025-11-17
>
> ## R4. Acknowledgement to previous dynamic benchmarks and clarification
>
> **Reviewer’s concern.**
> The “first” claim should be refined, should refer KORgym and Math-perturb.
>
> **Our response**
>
> We appreciate the your rigor but we are afraid some misunderstanding occurs. Here is our clarification.
>
> 1. **Scope clarification.**
>    Our intent was to claim that SciDA is the first scientific multi-domain, Olympiad-level and dynamic benchmark, which is like real scientific scenarios, to a certain extent.
>
> 2. **Relation to prior dynamic benchmarks.**
>    We deeply and sincerely acknowledge the prior dynamic benchmarks. Therefore, we have discussed and cited prior dynamic benchmarks (e.g., KORgym, Math-perturb, VarBench etc.) in Section 2.2.
>    - **KORgym** realizes dynamic initialization but focuses on game-based/interaction tasks and toy problems, not curated scientific Olympiad-style problems.
>    - **Math-perturb** targets only mathematics and does not cover physics, chemistry, or biology.
>
> ## R5. Figure 3 & 4
>
> We would like to elaborate it more clearly:
>
> 1. **Figure 3 (reasoning behaviors).**
>    In Section 5.1, under randomized parameters, models exhibit more reasoning behaviors (deep reasoning, self-exploration, self-reflection) for the same base problems, indicating that they cannot rely on memorized patterns and instead engage in longer, more exploratory chains of thought.
>
> 2. **Figure 4 (error types).**
>    Figure 4 then shows that, when more reasoning is involved, the majority of failures are calculation errors rather than logical errors in most subjects.
>
> Taken together, these results indicate that: models often do generalize the high-level logical template of a problem, but the full process of reasoning involves more (As discussed in R2).
>
> Again, we would like to thank you for the constructive feedback. We are glad to offer this clarification and we believe the above clarifications and adjustments (more explicit statistics, refined claims, clearer interpretation of error types and reasoning behavior) will strengthen the paper and better!

---

### Official Review · Reviewer_qfU1 · 2025-10-27

**Soundness:** 2
**Presentation:** 2
**Contribution:** 2
**Rating:** 2
**Confidence:** 4

**Summary:**

This paper presents SciDA (Scientific Dynamic Assessor of LLMs), a multidisciplinary dynamic benchmarking framework designed to comprehensively and authentically evaluate the scientific reasoning capabilities of large language models (LLMs). SciDA comprises over 1,000 olympiad-level numerical computation problems, each of which can be randomly initialized with different numerical values at every inference attempt, thereby preventing models from relying on fixed numerical patterns. Experiments conducted on SciDA with multiple top-tier open-source and closed-source LLMs reveal a significant performance drop under randomly initialized numerical values, demonstrating that SciDA provides a realistic and unbiased assessment of numerical reasoning abilities.

**Strengths:**

SciDA effectively prevents models from relying on fixed numerical patterns by randomly initializing the variables in each problem. Moreover, SciDA spans multiple disciplines, including mathematics, physics, chemistry, and biology, and all its problems are drawn from Olympiad-level competitions, ensuring high quality and complexity. This comprehensive evaluation approach provides researchers with a more realistic and holistic assessment tool for evaluating the scientific reasoning capabilities of large language models (LLMs).

**Weaknesses:**

- The core approach of SciDA, randomly initializing variables within problems, effectively mitigates model reliance on fixed numerical patterns. While this strategy reduces the risk of data contamination to some extent, it remains relatively simplistic and lacks deeper analysis or evaluation of the model’s actual reasoning process.
- Although SciDA’s dynamic initialization strategy addresses data contamination to a degree, similar techniques have already been employed in other domains, such as dynamic benchmarking. Consequently, SciDA does not represent a significant innovation in methodology.
- The approach may also lack flexibility: the range and manner of random initialization are fixed and not adaptively tuned based on problem difficulty or model capability. This rigidity could result in some problems becoming either too easy or excessively hard after randomization, thereby failing to accurately reflect the model’s true reasoning capacity.

**Questions:**

1. In the experiments involving random initialization, did the authors consider potential differences in how sensitive various models are to such randomization? For instance, some models might be more robust and adaptable to randomized inputs, while others could be heavily reliant on fixed numerical patterns.

2. Within the dynamic initialization strategy, did the authors explore the possibility of adaptively adjusting the range and distribution of variables based on problem difficulty and model capability? For example, for particularly challenging problems, could narrowing the variable range help moderate difficulty and thereby yield a more precise assessment of a model’s reasoning ability?

3. In the error analysis section, did the authors consider performing a finer-grained categorization of model errors? Beyond logical and computational errors, could additional error types—such as misinterpretation (understanding errors) or formulation/representation errors (expression errors)—be introduced to provide deeper insights into model failure modes?

---

> ### Author Response · Authors · 2025-11-17
>
> Thank you very much for your thoughtful review and valuable feedback. We sincerely appreciate the time and effort you have dedicated to assessing our work. Your comments are highly insightful and provide excellent guidance for strengthening our paper. Below, we provide a point-by-point response to the weaknesses and questions you have raised.
>
> **R1: The concern that our approach is simplistic and lacks analysis of the model's reasoning process**
>
> We agree that analyzing the model's reasoning behavior is crucial. In response to this, we dedicated Section 5.1 (lines 375-410) of our paper to this exact analysis. Our core finding is that the random initialization setting fosters more analytical and proactive problem-solving, requiring the model to engage in more in-depth reasoning, whereas the initial fixed-value setting often leads to quicker, more memory-based responses. We will clarify this point in the revised manuscript and are prepared to include more detailed analysis or additional experiments if you have specific suggestions on what aspects to explore further.
>
> **R2:  The perceived lack of methodological innovation**
>
> We sincerely acknowledge the prior work in dynamic benchmarking. Our paper already discusses and cites several relevant dynamic benchmarks (e.g., KORgym, Math-perturb, VarBench) in Section 2.2. However, constructing a dynamic benchmark for scientific problems presents unique and significant challenges. **To the best of our knowledge, SciDA represents the first effort to successfully build and apply a dynamic benchmark in the scientific domain.** Furthermore, the specific technique of using dynamically generated variables within scientific questions is a novel contribution of our work. We will revise the text to highlight this innovation more clearly.
>
> **R3: The flexibility of the randomization range and potential impact on problem difficulty**
>
> This is a valuable point. In our selection process, we specifically chose problems where the answer can be expressed as a function of its input variables (i.e., ans = func(x₁, x₂, …, xₙ)). This key criterion ensures that varying the numerical values within a reasonable range does not alter the fundamental logical structure or conceptual difficulty of the problem. The randomization tests the model's ability to execute the correct reasoning pathway with new inputs, not to change the pathway itself.
>
> **R4: differences in model sensitivity to randomization**
>
> This is an excellent point, and we thank the reviewer for suggesting a deeper analysis of model robustness. Our results in Table 1 indeed reveal significant variation in how sensitive different models are to randomized inputs.As the reviewer implies, a key metric for robustness is the performance gap between the Initial (fixed-parameter) and Random (dynamically parameterized) settings. A smaller gap indicates a more robust model that relies less on memorized numerical patterns and more on generalizable reasoning.A central finding from this analysis is that models equipped with a code interpreter (CI) consistently demonstrate the highest level of computational robustness. For instance:(1)Gemini-2.5-pro.preview.0506.google.ci shows an "All" score drop from 49.84 (Initial) to 38.26 (Random), a gap of ~11.6 points.(2)OpenAI-o3-high.code shows a similar high robustness, with a gap of ~15.3 points (52.22 to 36.90). In contrast, models without this capability often exhibit a much larger performance decrease. This strongly suggests that the ability to offload calculation to a formal, symbolic process (code) significantly enhances a model's stability against numerical variations. We will revise the manuscript to include a systematic analysis of these robustness gaps, explicitly discussing the performance of CI-enabled models versus others.
>
> **R5: Regarding adaptively adjusting the randomization range**
>
> In this work, our primary goal was to establish a fair and uniform benchmarking baseline across all models and problems. Employing a fixed randomization strategy was a deliberate choice to ensure that performance differences are directly comparable and not confounded by multiple, adaptive difficulty levels.
>
> We hope our responses have addressed your concerns thoroughly. We are glad to offer these clarifications and we hope this response finds you well. If you have other problems or you need more detailed information, feel free to contact us. Should you find our clarifications valuable, we would be grateful if you could reconsider your overall evaluation of our submission.

---

### Official Review · Reviewer_GamU · 2025-10-28

**Soundness:** 3
**Presentation:** 2
**Contribution:** 3
**Rating:** 6
**Confidence:** 3

**Summary:**

This paper introduces SciDA, a multidisciplinary benchmark designed to assess the scientific reasoning ability of LLMs under dynamic randomized conditions. The authors identify a critical issue about data contamination in the current benchmarks for evaluating LLMs. To address this, SciDA collects over 1K Olympiad-level scientific computation problems covering math, physics, chemistry, and biology, each parameterized with random variable ranges to eliminate memorization effects. The benchmarking approach involves expert curation and dynamic random initialization. Experiments on 14 mainstream LLMs reveal that accuracy drops by up to 60% under randomization, exposing an overestimation of current LLMs' reasoning ability. The paper concludes that SciDA offers a contamination-free approach for evaluating genuine reasoning performance, with plans for future expansion into more disciplines.

**Strengths:**

- The data curation pipeline with Olympiad-level problems and expert annotation ensures quality and complexity.

- The empirical findings of systematic performance drop show the effect of randomized conditions and reveal concerns of data contamination.

- Code is provided for reproducibility.

**Weaknesses:**

- Some statements are a bit overclaimed.
	- The claim of being "contamination-proof" is not fully substantiated. This is actually a very hard problem to completely solve.
	- In the abstract, there is "we provide truthful and unbiased assessments of the numerical reasoning capabilities of LLMs", but there are actually no guarantees.

- Writing can be refined for conciseness and professional tone.

**Questions:**

- How to ensure that the dynamic randomization does not inadvertently generate unsolvable problems beyond the predefined ranges?

- How to verify the "contamination-proof" claim? How to verify empirically that SciDA problems are unseen in major training corpora like Common Crawl?

---

> ### Author Response · Authors · 2025-11-17
>
> Thank you for your meticulous and careful review. Our team has carefully considered the questions you raised and our responses are as follows:
>
> **R1:How to ensure the dynamic randomization does not inadvertently generate unsolveable problems beyond the predefined ranges**
>
> This problem is one of our major concern. We solved this problem by two methods: (1) During the data annotation process, our team members manually annotate the meaningful range of each variable.Both the physical meaning and its meaning in the context of the question were considered, and our random initialization generates new questions based on this randomly generated variable range.（See Figure 1 Problem Paradigm;). (2) To verify whether our problems were solvable, we also manually wrote Python code to solve each problem(See Figure 2). If the code can produce a correct answer, then the problem is definitely computable.These two steps ensure that our data does not generate incalculable problems during random initialization, thus guaranteeing the authenticity and validity of the assessment data.
>
> **R2: Contamination-proof Verification Problem**
>
> The second question is about "contamination-proof" claim. **However, the fact that the large model has not seen the data is not our claim**. Our sources can be found at Appendix A, these data are likely to have been used during pre-training or post-training. After our random initialization, the model could no longer solve problems correctly even without any increase in difficulty, indicating that the model may have experienced data leakage during training. This is also the most important conclusion we drew from our data.
>
> At the same time, you list some of the weaknesses.
>
> **R3: Overclaim of some statements.**
>
> We thank the reviewer for this insightful comment. We agree that the terms "contamination-proof" and "guarantees" were too strong and could be misinterpreted. The central goal of our work is not to claim the complete solution to this very hard problem, but to **take a meaningful step forward by proposing a novel method to quantify the extent of data contamination**, which is a primary challenge in benchmarking. Our approach of randomizing variables in numerical problems provides, to our knowledge, the first quantitative lens for assessing contamination. Furthermore, it serves as an effective strategy to mitigate its effects and enhance model robustness. Again thank you for your advise, we will soften the claim in the abstract to "we provide a methodology for more truthful and unbiased assessments."
>
> Again we sincerely appreciate your positive feedback on our data quality and core findings, as well as the favorable rating. We hope our responses have addressed your concerns thoroughly. Should you find our clarifications valuable, we would be grateful if you could reconsider your overall evaluation of our submission.Thank you for your time and further consideration.

---

### Author Response · Authors · 2025-11-23
**Paper Updates**

Dear Area Chair and Reviewers,

We thank you again for your valuable feedback. We have now uploaded an updated version of the paper, which incorporates all requested clarifications and corrected typos present in the previous version. We hope these revisions address the concerns raised and further strengthen our submission.

Please let us know if any further information is needed. Looking forward to hearing from you.

Best regards, The authors

---

### Meta-Review · Area_Chair_a7iz · 2026-01-05

**Summary:**

The paper proposes SciDA, a dynamic benchmark of over 1,000 Olympiad-level numerical problems across mathematics, physics, chemistry, and biology. By randomizing input parameters for each inference, it aims to mitigate data contamination and provide a more truthful assessment of LLMs' scientific reasoning capabilities. Experiments show substantial performance drops under randomization for a range of top LLMs, suggesting overestimation in static benchmarks.

Reviewers recognized the importance of addressing data contamination, the quality of expert-curated problems, and insightful observations on numerical fragility. However, significant concerns emerged regarding methodological novelty, overclaimed contributions, interpretive ambiguity, and technical details. The randomization approach, while useful, builds directly on prior dynamic benchmarks (e.g., KORgym, Math-perturb, VarBench) without sufficient innovation for a multi-domain scientific setting. Claims of being "contamination-proof" or providing "unbiased" assessments were viewed as overstated. The main finding—that drops stem largely from calculation rather than logical errors—raises questions about whether the core issue is brittle arithmetic (potentially mitigated by tools) rather than absent reasoning. Dataset scale limits sub-domain reliability, and details on randomization ranges, correctness tolerances, and statistical uncertainty were initially insufficient.

The authors' rebuttal and paper updates softened claims, added citations to priors, clarified validation processes, refined interpretations of error types and reasoning behavior, and committed to including uncertainty metrics. These responses were constructive and addressed many specific points, but did not fundamentally resolve the core issues of limited novelty and impact relative to existing dynamic evaluation methods.

**Reviewer Concerns:**

Most technical concerns (e.g., solvability, range definition, tolerances) were adequately clarified or promised for revision. However, concerns about methodological originality, overclaiming, and the precise implications of calculation-dominant errors remain outstanding, as the core contribution is seen as an application/extension of established dynamic benchmarking rather than a substantial advance.

**Reviewer Scores:**

Initial scores were mixed: one clear reject (2), two marginally below acceptance (4), and one marginally above (6). The rebuttal provided useful clarifications but no reviewers indicated upward revisions. Post-discussion scores likely remain in the borderline-to-reject range.

---

### Decision · Program_Chairs · 2026-01-26

Reject